# Fluorescence-based thermal sensing with elastic organic crystals

Qi Di[1], Liang Li[2,3], Xiaodan Miao[4], Linfeng Lan[1], Xu Yu[1], Bin Liu[1], Yuanping Yi ●[4] ✉, Panče Naumov ●[2,5,6] ✉ & Hongyu Zhang ●[1] ✉

Operation of temperature sensors over extended temperature ranges, and particularly in extreme conditions, poses challenges with both the mechanical integrity of the sensing material and the operational range of the sensor. With an emissive bendable organic crystalline material, here we propose that organic crystals can be used as mechanically robust and compliant fluorescence-based thermal sensors with wide range of temperature coverage and complete retention of mechanical elasticity. The exemplary material described remains elastically bendable and shows highly linear correlation with the emission wavelength and intensity between 77 K to 277 K, while it also transduces its own fluorescence in active waveguiding mode. This universal new approach expands the materials available for optical thermal sensing to a vast number of organic crystals as a new class of engineering materials and opens opportunities for the design of lightweight, organic fluorescence-based thermal sensors that can operate under extreme temperature conditions such as are the ones that will be encountered in future space exploration missions.

Exploration of extreme environments, such as for example the possibility to sustain life at very low temperatures on Earth or in outer space requires robust cryogenic engineering materials that will be able to respond to a number of requirements[1–3]. Cooling has an inevitable, and sometimes drastic effect on the properties of both biogenic and artificial materials, as is well known with liquefaction or solidification of natural gas[4], deceleration or cessation of cell metabolism[5,6], and low-temperature superconductivity[7,8]. The oil and gas, machining, chemical industry, food, biomedicine, and aerospace sectors are in increasing demand for cryogenically stable and robust temperature sensing materials[9–11]. Low-temperature environments also imply the need for *accurate* measurement of low temperatures, which currently relies on either thermally controlled resistance[12] or integrated circuit (IC) temperature sensing[13]. However, devices based on resistive measurement may fail due to the opening of the electrical circuit by mechanical

separation of the components or as a result of materials' aging. While the IC sensors are comparatively more robust, they typically cover only a narrow temperature range for measurement. While being effective for specific applications, a common drawback of both of these sensing technologies is that their physical components are made of stiff and brittle materials, and are therefore prone to mechanical damage in devices that are exposed to vibrations or shock over prolonged periods of time. A long sought-after alternative to these materials are fundamentally new soft, light, mechanically compliant, and robust temperature-sensing materials that can be used for accurate and reliable thermal measurement.

Organic fluorescent crystals based on π-conjugated small organic molecules with photoluminescent properties have been reported[14]. In a crystalline condensed state, both the intermolecular interactions and the molecular arrangement are known to affect the luminescence[15–17].

[1]State Key Laboratory of Supramolecular Structure and Materials, College of Chemistry, Jilin University, 130012 Changchun, China. [2]Smart Materials Lab, New York University Abu Dhabi, PO Box 129188 Abu Dhabi, UAE. [3]Department of Sciences and Engineering, Sorbonne University Abu Dhabi, PO Box 38044 Abu Dhabi, UAE. [4]Beijing National Laboratory for Molecular Sciences, CAS Key Laboratory of Organic Solids, Institute of Chemistry, Chinese Academy of Sciences, 100190 Beijing, China. [5]Center for Smart Engineering Materials, New York University Abu Dhabi, PO Box 129188 Abu Dhabi, UAE. [6]Department of Chemistry, Molecular Design Institute, New York University, 100 Washington Square East, New York, NY 10003, USA. ✉e-mail: ypyi@iccas.ac.cn; pance.naumov@nyu.edu; hongyuzhang@jlu.edu.cn

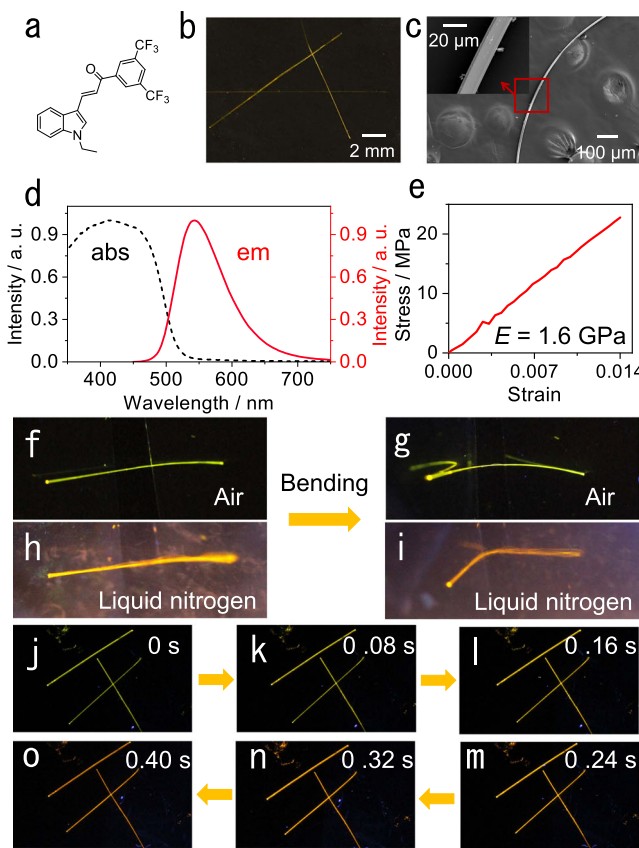

**Fig. 1 | Structure, elasticity, and fluorescence of compound 1. a** Chemical structure of compound 1. **b** Photograph of crystals of compound 1 under daylight. **c** SEM images of a bent crystal. The region in the boxed area is shown as a zoomed-in image in the inset. **d** Absorption (broken line) and emission (solid line) spectra of crystals of compound 1. **e** Stress-strain curve of the crystal. **f**, **g** Photographs showing crystal bending in air. **h**, **i** Photographs showing crystal bending in liquid nitrogen. **j**–**m** Change in emission color of the crystals from 298 K to 77 K (the images shown in panels **f**–**m** were recorded under UV light for better contrast).

Within that context, molecular organic crystals from π-conjugated molecules have been studied extensively as promising emissive materials due to the effects of molecular aggregation on the emission[18]. However, organic crystals have not been traditionally considered the materials of choice for temperature measurement. The recent realization of their chemical versatility, anisotropy in structure and properties, and long-range structural order has led to increased recognition of these compounds as a new materials class for organic optoelectronic components, such as resonators[19,20], circuits[21–24], and lasers[25,26]. A particularly important aspect of their property profile that has been highlighted only recently is their pronounced elasticity[27,28]. This newly realized facet is central to their mechanical compliance and has been recently explored for electronics applications that require lightweight materials, such as optical waveguides for passive transduction of information in both the visible and near-infrared spectral regions[29–33]. Opportunities for application of the fluorescence of some flexible organic crystals have also been implicated[34,35] for active signal transmission[36–39]. In line with the increasing demand for resilient cryogenic materials, we sought to develop lightweight, flexible materials that can be used for sensing or optical information transduction at low temperatures. The material that we report here combines all of these assets. Specifically, crystals of the organic crystalline material that we describe display reversible, reproducible, and strong temperature-induced shifts of their fluorescence, which translates into an opportunity for reproducible optical temperature measurement based on fluorescence. Within the broader set of available

luminescence thermometry techniques[40], this work brings a new class of materials that could be considered as an alternative to the currently used fluorescent thermometers.

## Results and discussion
### Preparation and characterization
The material described here, (*E*)−1-(3,5-bis(trifluoromethyl)phenyl)−3-(1-ethyl-1*H*-indol-3-yl)prop-2-en-1-one (compound 1; Fig. 1a), was discovered during our broader screening of materials in an effort to identify organic crystals that are both elastic and emissive. It was synthesized in two steps with a 72% yield (Supplementary Figs. 1 and 2). Long, yellow needle-like crystals of about 1.5 cm in length were readily prepared by liquid-phase diffusion (Fig. 1b). The material has a good thermal stability and does not undergo a phase transition up to its melting point at 467 K (Supplementary Fig. 3). The crystals of compound 1 are reversibly elastic; they can be bent without any visible damage (Supplementary Movie 1), as it was confirmed by the scanning electron microscopic images of bent specimens that were fixed in their bent state for inspection (Fig. 1c). No obvious defects could be observed on the crystal surface even after bending the crystals up to 5000 times (Supplementary Fig. 4). The crystals absorb light with a maximum at 415 nm, and emit bright green fluorescence with a maximum emission at about 540 nm and a quantum yield of 0.16 at 298 K (Fig. 1d). The fluorescence quantum yield is 0.30 at 77 K, which is higher compared to room temperature due to the decreased non-radiative rate at low temperature (Supplementary Table 1). Their bulk elastic modulus at 298 K was found to be 1.6 GPa by tensile measurements (Fig. 1e)[41]. The elasticity of the crystals is preserved even at low temperatures, and they can be bent repeatedly even when they are immersed in liquid nitrogen (Fig. 1f–i).

### Temperature-induced change of fluorescence
During the mechanical characterization at low temperature, we noticed that the color of emission of crystals of compound 1 changes visibly from green to orange when they are transferred from room temperature to liquid nitrogen (Fig. 1j–o; Supplementary Movie 2). The temperature-dependent fluorescence spectra show that the color change is due to a red-shift of the emission maximum upon cooling, whereupon the emission intensity gradually increases (Fig. 2a; Supplementary Fig. 5). The maximum emission wavelength changes from 540 to 580 nm upon cooling from 277 to 77 K, and the dependence is linear in the temperature range from 77 to 277 K ($\lambda_{max}$/nm = −0.21 $T$/K + 596.91, $R^2$ = 0.9948; Fig. 2b). More quantitatively, when plotted on a CIE 1931 color space diagram[42] this change in color of the emitted light is reflected in the change of the CIE coordinates from (0.38, 0.59) to (0.51, 0.48) (Fig. 2c). At 77 K, the emission intensity increases to 1.79 times compared to that at 277 K. The change of emission intensity with temperature is also linear within a certain temperature range ($I/I_{max}$ = −0.0022 $T$/K + 1.14, $R^2$ = 0.9711; Fig. 2d). This linear and strong response of the fluorescence of compound 1 with temperature favors this material as a temperature-sensing medium. The lowest detectable temperature of compound 1 is lower than that of some metal sensors[43,44], however, the sensitivity (maximum emission wavelength vs. temperature) is higher than other organic materials with temperature-dependent fluorescence[45,46]. In order to verify whether the red-shift of compound 1 is caused by triplet luminescence, the lifetimes at 298 and 77 K were determined (Supplementary Fig. 6). The lifetime of the emission of compound 1 at 77 K is 17.99 ns, which is longer than that at 298 K (6.13 ns), confirming that the emission does not originate from a triplet state. The lifetime and quantum yield of bent crystals were also recorded and indicate that the bending does not have any significant effect on non-radiative pathways (Supplementary Fig. 6, Supplementary Table 2).

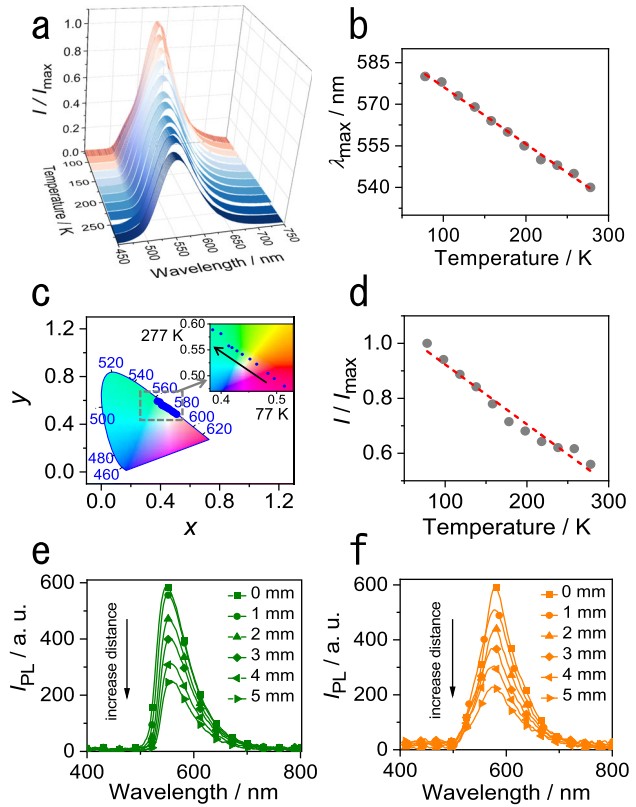

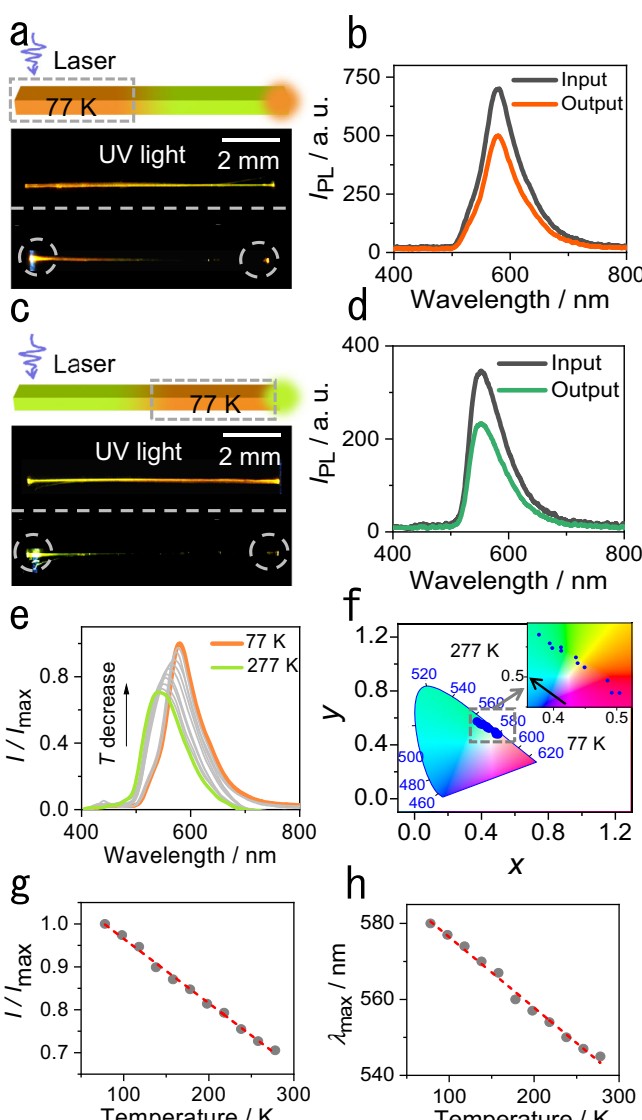

**Fig. 2 | Temperature dependence of the emission of compound 1. a** Variable-temperature emission spectra of a crystal of compound 1. **b** Correlation between crystal emission wavelength and temperature in the temperature range 77–277 K. The straight broken line shows a linear fit to the data. **c** The CIE coordinates of crystal emission bands at different temperatures. The region in the boxed area is shown as a zoomed-in image in the inset. **d** Correlation between the crystal fluorescence intensity ratio and temperature in the temperature range 77–277 K. The straight broken line shows a linear fit to the data. **e, f** Emission spectra were measured at one end of the crystal at 298 K (**e**) and 77 K (**f**). The values 0–5 mm, corresponding to curves with different symbols, represent the distance from the excitation site to the point of measurement. The excitation was performed at 355 nm.

## Optical-waveguiding capability

The remarkable flexibility of the compound 1 crystals at both ambient and low temperatures appears as a prospective platform for their application in thermally resistant (opto)electronics. Similar to other flexible crystals[47,48], single crystals were tested as active optical waveguides (Fig. 2e). The optical loss coefficient of a straight crystal at room temperature was found to be 0.16 dB mm⁻¹ (Supplementary Fig. 7). A bent crystal had a nearly identical optical loss, 0.17 dB mm⁻¹ (Supplementary Fig. 8). This result indicates that the crystal bending does not have a significant effect on the optical waveguiding performance, as is expected from the retention of the crystal integrity upon bending, described above. The cooling did not affect the waveguiding ability of the crystal (Fig. 2f); at 77 K, the optical loss factors were found to be 0.17 dB mm⁻¹ for a straight crystal and 0.20 dB mm⁻¹ for a bent crystal (Supplementary Figs. 9 and 10, Supplementary Movie 3).

## Temperature sensing using fluorescence

We hypothesized that the combination of linear dependence of the emission wavelength on the temperature and the favorable mechanical flexibility of compound 1 at low temperature could carry some potential for the development of flexible temperature sensors. Along this line of thought, a crystal of about 1.0 cm in length was selected, and liquid nitrogen was dropped continuously on a small area at one end (Fig. 3a). The cold spot was excited with a laser, and the optical

**Fig. 3 | Temperature-dependent optical waveguiding. a** Schematic diagram and real picture of a crystal waveguide at low temperature. The gray dashed box denotes the area where the liquid nitrogen was dropped (77 K). **b** Emission spectra recorded at the excitation end (77 K) and at the output end of the crystal (230 K). **c** Schematic diagram and real picture of a crystal waveguide at room temperature. **d** Emission spectra recorded at the excitation end (230 K) and at the output end of the crystal (77 K). **e** Emission spectra of the crystal optical waveguide at different temperatures. **f** CIE 1931 color space coordinates of crystal optical waveguide signals at different temperatures. The region in the boxed area is shown as a zoomed-in image in the inset. **g** Correlation between the emission intensity ratio and temperature in the temperature range 77–277 K. The straight broken line shows a linear fit to the data. **h** Correlation between the crystal waveguide emission wavelength and temperature in the temperature range 77–277 K. The straight broken line shows a linear fit to the data.

output was collected and analyzed at both ends of the crystal (Supplementary Fig. 11). Consistent with the temperature difference, the cold end of the crystal emitted orange light, while the opposite end that was at higher temperature emitted green light. The temperature of the warm end was determined to be about 230 K (Supplementary Figs. 12 and 13). However, the emission wavelength maximum of the spectra recorded at both ends was 580 nm, showing that only the orange light was transduced (Fig. 3b). We note that the optical waveguide is in active mode here. The photoluminescence (PL) spectra of crystals of millimeter size at 77 and 298 K were also recorded.

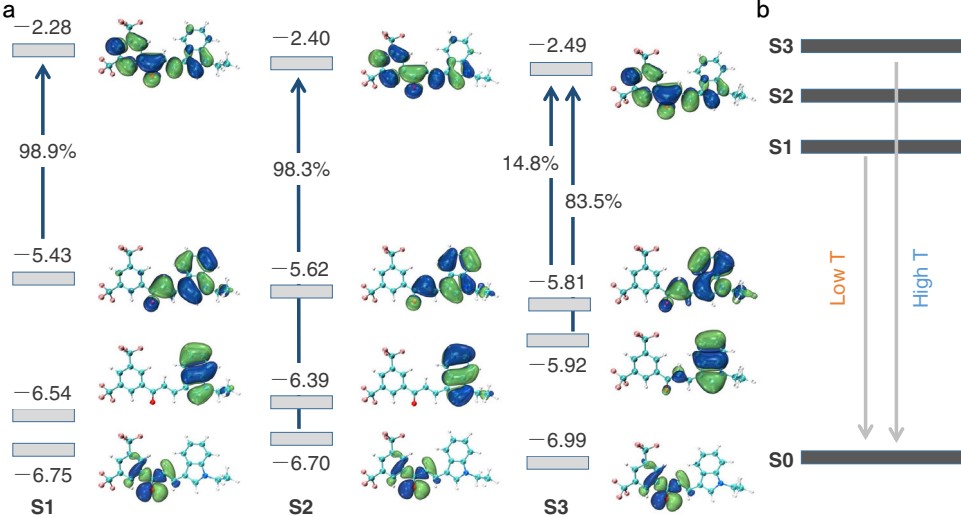

**Fig. 4 | Molecular orbitals related to the temperature-dependent emission.**
**a** Frontier orbitals and transition contributions for $S_1$, $S_2$ and $S_3$ states in the respective optimized geometries (energy unit: eV). **b** Energy level diagram showing the changes in the emission energy and the accompanying blueshift of the emission with increasing temperature caused by the thermal population of the higher-energy $S_3$ state.

Comparing the PL spectrum at 77 K with the optical waveguide signal at 77 K, we conclude that the peak shapes are almost identical (Supplementary Fig. 14d). This result indicates that the optical signal propagating in the waveguide is not affected during its propagation. Other two crystals that have been reported earlier (compounds 2 and 3) were also investigated to confirm this property of organic crystals (Supplementary Fig. 15). The position of laser excitation was then changed, and the crystal was excited at the higher temperature end, while it was being cooled at the opposite end (Fig. 3c). Contrary to the previous case, the warm end emitted green light and the cold end emitted orange light. The spectra of the fluorescence remained identical at both ends, with a maximum emission at 550 nm (Fig. 3d). These results demonstrate that the crystal can be indeed cooled locally, and it does not equilibrate thermally during the experiment. More importantly, when the crystal is used as a medium for optical transduction, the output signal depends only on the temperature at the point of excitation; the output is not affected even when the intermediate section of the crystal is at lower temperature.

In another experiment, a 1.5 cm-long crystal was selected, and one of its ends was brought into contact with evaporating liquid nitrogen. This cool end was excited with a laser at 355 nm, and the spectrum of the output light was recorded at the other end (Supplementary Fig. 16). As the liquid nitrogen evaporated, the excited end of the crystal gradually warmed up, and the output spectrum changed (Fig. 3e). As expected from the other experiments above, upon warming the position of the maximum emission was blueshifted while its intensity decreased, as it is illustrated by changes in the respective CIE coordinates (Fig. 3f). This is in accordance with the results obtained by cooling, as shown in Fig. 3g and h. The changes in maximum emission wavelength and intensity with temperature on warming are linear ($\lambda_{max}$/nm = −0.19 $T$/K + 595.00, $R^2$ = 0.9921; $I/I_{max}$ = −0.0015 $T$/K + 1.12, $R^2$ = 0.9964). These experiments demonstrate that the crystal does not only respond to temperature changes by emitting light, but it also transduces and outputs the temperature information by acting as an optical waveguide. The measurement was also performed after the crystal was repeatedly bent, and the results show that the sensitivity was practically unaffected even after the crystal was bent 1000 times (Supplementary Fig. 17, Supplementary Table 3). The signal from the crystal is not affected by the temperature of the section through which the optical signal is transmitted. The crystal's output reflects only the temperature of the point of excitation, and this property favors this material, and in principle also other emissive elastic organic crystals, as an active sensing medium for luminescence thermometry.

## Computational modeling of the fluorescence

To investigate the root cause of the temperature dependence of fluorescence, density functional theory (DFT) calculations were performed on a molecule of compound 1. Supplementary Fig. 18 shows the frontier molecular orbitals based on the ground-state ($S_0$) geometry of compound 1. Obviously, the LUMO, HOMO, and HOMO-1 are mostly π-orbitals, while the HOMO-2 is an n-orbital and distributed over the carbonyl group and its surroundings. The HOMO-1 is localized on the indole ring and the HOMO spreads over the indole ring and the adjacent vinyl group. In contrast, the LUMO is delocalized over the entire molecular backbone. Vertical excitations from $S_0$ to $S_1$, $S_2$, and $S_3$ are dominated by the transitions from the HOMO, HOMO-2, and HOMO-1 to LUMO, respectively. The $S_2$ and $S_3$ excitation energies are close to the $S_1$ excitation energy (Supplementary Table 4). Deeper insight into the fluorescence properties necessitated closer inspection of the electronic structures in the excited-state optimized geometries. The nature of the frontier orbitals in the excited state is hardly altered; moreover, the $S_1$ and $S_2$ excitations still correspond to the HOMO and HOMO-2 to LUMO transitions, respectively. Interestingly, in the $S_3$ geometry, besides the HOMO-1 → LUMO transition, the HOMO → LUMO transition has a significant contribution (14.8%) to the $S_3$ excitation, with significant oscillator strength ($f \approx 0.30$) (Fig. 4a). Because of the relatively close energies between the $S_1$ and $S_3$ states (Supplementary Tables 4 and 5), the $S_3$ population will be increased when the temperature is elevated, which is beneficial to enhance the luminescence from $S_3$ and results in blue-shift in the emission (Fig. 4b). This result supports the experimental observations.

## Relation to the crystal structure

The change in emissive properties of any crystal is also inevitably related to its crystal structure, and therefore the structure was determined at 298 and 100 K (Supplementary Fig. 19, Supplementary Table 6, Supplementary Data 1 and 2). At 298 K, the crystal is in the monoclinic system (space group $P2_1/n$) with $Z = 4$ and cell parameters $a$ = 12.1007(9) Å, $b$ = 4.8876(4) Å, $c$ = 31.714(2) Å, and $\beta$ = 91.548(3)°, and unit cell volume $V$ = 1875.0(2) Å$^3$. The torsion angles around the two

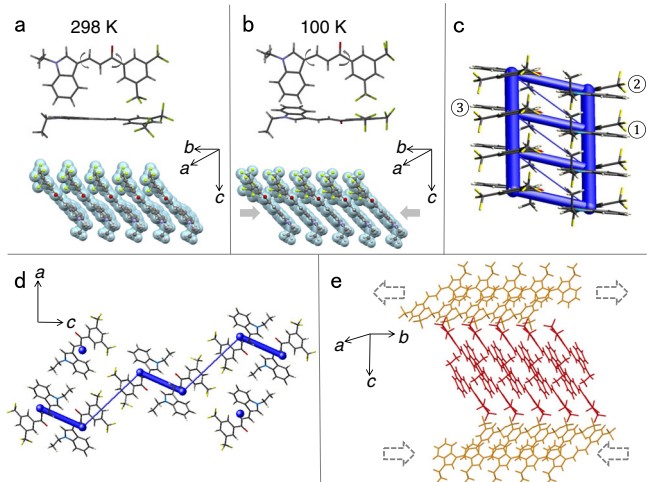

**Fig. 5 | Effect of temperature on the structure and energy framework analysis of compound 1. a, b** Molecular geometry and packing at 298 K (**a**) and 100 K (**b**). The arrows indicate the torsional angles discussed in the text. **c** Energy bars showing the interaction energies. The blue bars represent the interactions between molecules, and their thickness indicates the value of interaction energy. **d** Energy frameworks viewed along the *b* axis are consistent with the qualitative requirements for bending. **e** Parallel packing structure, and the purported expansion and contraction of the outer and inner arcs of the crystal that are suggested to occur during the bending of the crystal.

single chemical bonds between the benzene and the indole rings are 12.2° and 1.3° (Supplementary Fig. 19). The molecules are stacked in a parallel fashion and form columnar structures by π···π interactions with an intermolecular distance of 3.758 Å along the crystallographic [010] direction (Fig. 5a). Upon cooling to 100 K, the crystal symmetry is retained, but the cell parameters change to $a = 12.0224(7)$ Å, $b = 4.8149(3)$ Å, $c = 31.4650(18)$ Å and $\beta = 92.271(2)°$, and the unit volume is $V = 1819.97(19)$ Å³. The unit cell shrinks by 0.7% along the *a* axis, 1.5% along the *b* axis, and 0.8% along the *c* axis. The torsion angles between the benzene ring and the indole ring remain very close, 11.0° and 1.4° respectively, and thus the high degree of planarity of the molecule is retained at low temperature (Supplementary Fig. 19). The stacking distance of the molecules is slightly reduced, to 3.579 Å (Fig. 5b). Since the molecular geometry is practically conserved, similar to other cases[49,50], the change in emission can be attributed to the decreased π···π distance, which is related to the significant shrinking of the lattice along the *b* axis. Furthermore, the enhanced intensity of emission at low temperatures is related to the restriction of non-radiative transition pathways and enhancement of the competing radiative processes[51–53]. In addition, the interaction energy among the molecules at 298 and 100 K were both calculated. The energy frameworks were constructed by using Crystal Explorer[54] and the B3LYP hybrid functional with the 6–31G(d,p) basis set, where semiempirical dispersion was included by using D2 version of Grimme's dispersion, and they are visualized in Fig. 5c. Based on the preceding discussion, the structural changes observed upon cooling indicate that the unit cell contraction along the *b* axis is the most likely contributor to the appealing change in emission. The calculated interaction energy between molecule 1 and molecule 2 of –66.4 kJ mol⁻¹ at room temperature was found to decrease to –67.5 kJ mol⁻¹ at 100 K (Table 1). This decrease in energy is in accord with the strengthening of the π···π interaction and is also consistent with the observations from the crystal structures.

The crystal structure is also directly related to the elastic properties of compound 1. The thicker blue bars in Fig. 5c correspond to the π···π interactions along the [010] direction. The C–H···O interactions ($d = 2.600$ Å, $\theta = 148.8°$ and $d = 2.418$ Å, $\theta = 135.6°$) and F···F interactions

($d = 2.870$ Å) among the molecular layers in the *c* direction can also be identified in the energy frameworks, while the thinner blue bars among the layers correspond to the F···F interactions (Fig. 5d). As explained above, the interactions along the *c* axis are likely responsible for the extraordinary flexibility of the crystal. The intermolecular interactions along the *a* axis participate in the 'bendable' (100) plane (Supplementary Fig. 20). These supramolecular interactions could effectively absorb and dissipate the applied stresses, thereby rendering the crystal more susceptible to stress. In elastically bendable crystals, the stretching of the outer arc of the crystal and separation of the respective molecules, and the concomitant contraction of the inner layer and decrease of the distance between the molecules are thought to ensure reversibility of structural changes upon bending, that is, the elasticity of the crystal (Fig. 5e)[55–66]. Although the thermal changes in the structure cannot be taken as an indication of the structural changes that actually occur during bending[67], upon cooling these interactions in compound 1 are maintained with only a slight decrease in the respective distances (F···F, $d = 2.861$ Å; C–H···O, $d = 2.532$ Å, $\theta = 131.8°$ and $d = 2.335$ Å, $\theta = 140.0°$) due to the nearly isotropic contraction of the crystal. The result is also consistent with the slight decrease of intermolecular interaction energy at 100 K compared with 298 K (Table 1). This retention of overall interactions might be the reason for the retention of the compliant mechanical properties at low temperature.

In summary, a flexible organic crystalline material is reported here that was found to exhibit temperature-dependent spectral changes which could be used for measurement of temperature by optical means in a wide temperature range, particularly at low temperatures. The fluorescence of the material displays linearity in response with respect to both the maximum and intensity of the emission. Variable-temperature experiments on the optical wave-guides properties further demonstrated that these organic crystals can convert the temperature at the excited position into a more stable optical signal output. The extraordinary elasticity of the crystal is a desirable property, as it is expected to translate into better durability and resistance to mechanical damage when it is being used as an optical waveguide. An attempt was made to rationalize the temperature dependence of the emission and the elastic properties of the crystal by theoretical calculations and analysis of its crystal structure. The thermal population of a higher-lying S₃ state was identified as the most likely factor behind the change in emission energy. This work expands the scope of application of elastic organic crystals as optical waveguides at low temperatures and provides an alternative route to the development of fluorescence thermometric devices by using soft and light organic crystalline materials as active sensing medium.

## Methods

### Synthetic procedure for compound 1

All chemicals for synthesis were purchased from commercial sources. The solvents were analytical-reagent grade and were used without further purification. For the synthesis (Supplementary Fig. 21), 1*H*-indole-3-carbaldehyde (1.45 g, 10 mmol) was dissolved in 50 mL acetone, and $K_2CO_3$ (4.15 g, 30 mmol) was added. Then 1.5 mL $ICH_2CH_3$ was injected into the mixture, and the mixture was refluxed for 4 h. After cooling to room temperature, the $K_2CO_3$ was filtered out, and the solvent was removed by distillation under reduced pressure. 80 mL ethanol, 1-(3,5-bis(trifluoromethyl)phenyl)ethan-1-one (2.56 g, 10 mmol) and NaOH (0.08 g, 2 mmol) were added, and the mixture was refluxed for 4 h. The generated precipitate was filtered and washed with ethanol to give the crude product, which was purified by vacuum sublimation to produce compound 1 as a yellow solid (2.96 g, 72% yield). ¹H NMR (500 MHz, DMSO-d₆) δ 8.67 (d, *J* = 1.7 Hz, 2 H), 8.39 (s, 1 H), 8.32 (s, 1 H), 8.23–8.08 (m, 2 H), 7.73 (d, *J* = 15.3 Hz, 1 H), 7.67–7.55 (m, 1 H), 7.31 (dtd, *J* = 15.7, 7.2, 1.3 Hz, 2 H), 4.30 (q, *J* = 7.2 Hz, 2 H), 1.44

**Table 1 | Intermolecular interaction energy of compound 1**

| Temperature (K) | Method | $E_{total}$(1–2) (kJ mol$^{-1}$) | $E_{total}$(1–3) (kJ mol$^{-1}$) |
|---|---|---|---|
| 100 | B3LYP/ 6-31 G(d,p) | –67.5 | –52.4 |
| 298 | B3LYP/ 6-31 G(d,p) | –66.4 | –47.4 |

The intermolecular interaction energy between different molecules (molecules 1–3, as shown in Fig. 5c).

(t, $J = 7.3$ Hz, 3 H). $^{13}$C NMR (126 MHz, DMSO-d$_6$) $\delta$ 186.62, 141.09, 140.80, 137.51, 136.06, 131.35, 131.09, 129.04, 126.50, 125.98, 124.74, 123.41, 122.57, 122.09, 121.04, 114.78, 112.61, 111.43, 41.53, 15.56.

## Characterization

The $^{1}$H and $^{13}$C{$^{1}$H} NMR spectra were recorded on a Bruker Avance 500 MHz spectrometer with tetramethylsilane as an internal standard. The UV–vis absorption spectra were recorded with a Shimadzu UV-2550 spectrophotometer. The mechanical tests were carried out on no. 5944 Universal Testing System from Instron. The emission spectra were recorded with a Shimadzu RF-5301PC spectrometer or with an Ocean Insight Maya2000 Pro spectrometer. Differential scanning calorimetric (DSC) measurements were performed on a TA Instruments DSC Q20 calorimeter. For the optical measurements, the crystal was irradiated with the third harmonic (355 nm) of a Nd:YAG (yttrium–aluminum–garnet) laser at a repetition rate of 10 Hz and pulse duration of about 10 ns. The energy of the laser was adjusted by using the calibrated neutral density filters. The beam was focused into a stripe whose shape was adjusted to $3.3 \times 0.6$ mm by using a cylindrical lens and a slit. The emission was detected at one end of the crystal using a Maya2000 Pro CCD spectrometer. The temperature was measured by using a Miaoxin T10R-PT temperature sensor with 0.1 K resolution (reading error: ±0.3 K). The fluorescence quantum yields were determined by using a Hamamatsu Quantaurus-QY spectrometer. The light source (365 nm) was first used to excite a blank sample to obtain the background spectrum, and then the sample was placed in the sample pool to record its fluorescence spectrum. The integral difference between the two spectral excitation ranges was denoted as S1 and the integral sample emission range was denoted as S2. The quantum yield was calculated as the ratio of S2 to S1. The low temperature for the quantum yield measurement was obtained by using liquid nitrogen.

## Computational details

All DFT and TDDFT calculations were carried out by using the Gaussian 16 package[68]. The ground state and excited state geometries for the molecule of compound 1 were optimized at the DFT/B3LYP/6-31G(d,p) and TDDFT/M062X/TZVP levels, respectively. Based on the optimized geometries, the excited-state properties were calculated at the TDDFT/B3LYP/6-31G(d,p) level. In all calculations, the environment was modeled by using the polarizable continuum model (PCM) with dichloromethane ($\varepsilon = 8.93$) as solvent.

## Single crystal X-ray diffraction

The single-crystal X-ray diffraction data were collected on a Rigaku Synergy-DS Diffractometer. The data collection, integration, scaling, and absorption corrections were performed by using the Bruker Apex 3 software[69]. The structures were solved with direct methods using Olex2[70] and refined by using the full-matrix least-squares method on $F^2$. The non-hydrogen atoms were refined anisotropically. The positions of the hydrogen atoms were calculated and refined isotropically. The program PLATON was used for the geometrical calculations[71]. The graphics related to the structures were generated by using Mercury 4.2.0[72]. The crystallographic details are available in Supplementary Data 1 and 2.

## Data availability

Crystallographic data for the structures reported in this Article have been deposited at the Cambridge Crystallographic Data Centre, under deposition numbers CCDC 2155928 and 2155929. These data can be obtained free of charge from The Cambridge Crystallographic Data Centre via www.ccdc.cam.ac.uk/data_request/cif. All data are available from the corresponding authors upon request.

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

## Acknowledgements
This work was supported by the National Natural Science Foundation of China (52173164, H.Z.) and a fund from New York University Abu Dhabi (P.N.). The authors thank Ying Wang from the Technical Institute of Physics and Chemistry, Chinese Academy of Sciences.

## Author contributions
Q.D., P.N., and H.Z. conceived the study. Q.D. prepared and performed most experiments on the crystals. Q.D. and B.L. performed a series of experiments on crystal optical waveguiding properties, X.M. and Y.Y. performed the density functional theory calculations. Q.D., X.Y., and L.F.L. revised the article under the guidance of L.L. The paper was written with contributions from Q.D., L.L., P.N., and H.Z. All authors have given their approval for the final version of this paper.

## Competing interests
The authors declare no competing interests.
