## [Peer Review File · Nature Communications]

Fluorescence-Based Thermal Sensing with Elastic Organic CrystalsREVIEWER COMMENTS

Reviewer #1 (Remarks to the Author):

This is a well written paper that reports a crystalline material which has a linear emission profile with respect to temperature. The crystals grow in a habit which also allows them to be elastically bent and thus display wave-guiding properties. The findings are interesting, scientifically sound and well performed. The manuscript is publishable after minor proof-reading for grammar and style. It does not require further review.

I found it difficult to review the paper with the figures at the end. Additionally I would strongly recommend calling the compound 1 rather than BPEPO.

Reviewer #2 (Remarks to the Author):

Comments to the authors

Fluorescence-based Thermal Sensing with Elastic Organic Crystal. It is an exciting paper. I recommend it for publication after the authors address the following suggestions.

In-Page 2, line 46 for circuits, I suggest the authors cite Chemical Communications, 2022, 58 (21), 3415-3428 and Small, 2021, 17 (24), 2100277.

In-Page 2, line 50, original literature needs to be cited for passive transduction. Advanced Optical Materials, 2013, 1 (4), 305-311

"BPEPO are reversibly elastic; they can be bent without any visible damage, as was confirmed by inspecting the scanning electron microscopy of bent specimens". Why is the crystal being elastic not demonstrating its elasticity and permanently deformed as proven by SEM?

The crystal length is 1 cm; what is the size of liquid nitrogen that was dropped continuously on a small area at one end?

Fig 3. Provide the scale for images.

How to ascertain that the crystal (length, 1.5 cm) temp is 77 K at one end and the other end is 230 K? What is the error in the reading? How was the temp measured in such a localized manner? Include a detailed scheme for the experimental set-up and the temperature sensors' specification.

Page 4: Though orange (cold) and green (warm) FL emitted at opposite crystal terminals, the observation of 580 nm peak, showing that only the orange light was transduced, needs to be

reinterpreted. The spectral band shape and width are critical here. Fig. 3b spectrum is more symmetric (with a shoulder at about 520 nm), and Fig 3d is asymmetric (more intense on the left side). It looks like the low-intensity green signal is buried within the highly intense orange signal spectrum. This result shows that the orange and green signals are mixed inside the crystals, as evident from Fig. 3e spectral shoulder at intermediate temperatures. A similar interpretation also applies to the 550 nm signal.

Reviewer #3 (Remarks to the Author):

This paper describes an interesting concept for fluorescence-based thermal sensing using elastic organic crystals. The reported crystalline material [(E)-1-(3,5-bis(trifluoromethyl)phenyl)-3-(1-ethyl-1H-indol-3-yl)prop-2-en-1-one (BPEPO)] exhibits an impressively linear correlation between the emission wavelength and fluorescence intensity between 77K and 277K. Density functional theory provides detail rationale for the observed behavior. This observation forms the foundation for a new thermal sensing mechanism and would be a valuable finding to share with the general scientific community of this journal. I recommend the authors consider the following points to improved their work:

1. The abstract mentions space exploration missions as a potential application for these fluorescence-based thermal sensors. Characterization about the lifetime of temperature sensing after repeated bending of one crystal would be valuable to connect back to this application. How is the sensitivity of BPEPO impacted after repeated bending? Has the lifetime of the flexibility of BPEPO been tested?
2. The introduction lacks background on the connection between fluorescence and organic crystals. This section would benefit from information about the molecular design of emissive crystals and aggregate-induced fluorescence. Adding in this information may also help contextualize the process that led to the discovery of BPEPO as an elastic emissive crystal, mentioned on page 2 line 66.
3. Page 2, lines 52-55, the structure of this sentence is difficult to read.
4. Page 3, line 75, was the reported bulk elastic modulus measured at room temperature? Was the bulk elastic modulus measured at lower temperatures?
5. The sensitivity of BPEPO seems to be reported as the slope of the intensity vs temperature curve (Page 3, lines 86 and 90). How does this compare to the sensitivity of other thermal sensors?
6. Page 4, lines 124-125, why is the output not affected when the intermediate section of the crystal is at a lower temperature? Is this common for crystal waveguides?
7. Page 6, lines 181-182, the claim about the non-radiative and radiative pathways at low temperature could be strengthened by calculating the non-radiative and radiative rate constants at 77K and 277K from the fluorescence lifetime data.
8. Page 6, lines 199-200, does the mechanism that allows for the flexibility of BPEPO also promote the non-radiative pathways at room temperature? Is there a reduction of flexibility that corresponds to emission enhancement?

9. Does the fluorescent emission of BPEPO change at warm temperature (>277K)? If so, is this material as sensitive to warm temperature changes as it is towards cold temperature changes?
10. In Figure 1, it's difficult to make out the details in images h. and i.
11. The caption of Figure 1 mentions image b. depicts BPEPO under daylight. Are the other images also captured under daylight?
12. In the caption of Figure 2, the description of the locations on the crystal is unclear under part e,f.
13. In Figure 3, part a and c, what is the dashed grey box denoting?
14. In Figure 4, part b, are the colors of the S1, S2, and S3 levels meant to communicate information about emission? If so, that should be mentioned in the figure caption. If not, consider using greyscale for these levels.
15. Supporting Information should include cartesian coordinate files of optimized geometry for all structures.
16. The ramp rate used for DSC measurements should be reported in Figure S4 caption or in General Information section of the Supporting Information.
17. The method used to calculate the quantum yield should be detailed in the General Information section of the Supporting Information.

RESPONSE TO THE REVIEWERS' COMMENTS

In the PDF version of this response, the reviewers' comments are written in *black italic font*, our responses to the comments are written in **blue regular font**, and the changes made to the main text are written in **red regular font**.

Reviewer #1

Comment: *This is a well written paper that reports a crystalline material which has a linear emission profile with respect to temperature. The crystals grow in a habit which also allows them to be elastically bent and thus display wave-guiding properties. The finding are interesting, scientifically sound and well performed. The manuscript is publishable after minor proof-reading for grammar and style. It does not require further review.*

I found it difficult to review the paper with the figures at the end. Additionally I would strongly recommend calling the compound 1 rather than BPEPO.

Response: We thank the Reviewer for reading the manuscript and the comments. We agree with the Reviewer that it is difficult to read the manuscript with the figures at the end of the manuscript. We have indeed considered incorporating the figures in the main text close to the point where they are referred to for the first time, however we had to follow the guidelines of the journal (Nature Communications) which suggests that the figures be placed at the end of the manuscript.

Following the Reviewer's suggestion, the acronym "BPEPO" in the main text was replaced with "compound 1".

Reviewer #2

Comments to the authors

Fluorescence-based Thermal Sensing with Elastic Organic Crystal. It is an exciting paper. I recommend it for publication after the authors address the following suggestions.

Comment 1: *In-Page 2, line 46 for circuits, I suggest the authors cite Chemical Communications, 2022, 58 (21), 3415-3428 and Small, 2021, 17 (24), 2100277.*

Response: We thank the Reviewer for the careful reading and the comments. In the revised version of the manuscript, the suggested references are cited on page 2, line 51 (as References [21] and [22]).

Added References:

[21] Chandrasekar, R. Mechanophotonics-a guide to integrating microcrystals toward monolithic and hybrid all-organic photonic circuits. *Chem. Commun.*, **58**, 3415–3428 (2022)

[22] Chandrasekar, R. Mechanophotonics-mechanical micromanipulation of single-crystals toward organic photonic integrated circuits. *Small*. **17**, 2100277 (2021).

Comment 2: *In-Page 2, line 50, original literature needs to be cited for passive transduction. Advanced Optical Materials, 2013, 1 (4), 305-311*

Response: The reference is added on page 2, line 56, as Reference [29].

Added Reference:

[29] Chandrasekhar, N., Mohiddon, M. & Chandrasekar, R. Organic submicro tubular optical waveguides: Self-assembly, diverse geometries, efficiency, and remote sensing properties. *Adv. Optical Mater.* **1**, 305–311 (2013).

Comment 3: *"BPEPO are reversibly elastic; they can be bent without any visible damage, as was confirmed by inspecting the scanning electron microscopy of bent specimens". Why is the crystal being elastic not demonstrating its elasticity and permanently deformed as proven by SEM?*

Response: We thank the Reviewer for this comment. We believe that the way this was formulated has led the Reviewer to think that the crystal was permanently deformed under SEM. In fact, the crystal that was used for SEM was attached to the sample plate with conductive adhesive in its bent state for purpose of inspection. The SEM images, indeed, showed that there is no damage or defects on the surface of the bent crystal. In order to confirm that there is no damage after repeated bending, we performed additional experiments. A crystal was bent many times and observed using a microscope (Supplementary Movie 2). The optical images of the crystal that was bent 50, 100, 300, and 500 times are shown below (Figure R1), and they did not show any signs of deterioration of the crystal. In the revised version of the manuscript, the following sentence was modified to clarify this point (page 3, line 76):

"The crystals of compound 1 are reversibly elastic; they can be bent without any visible damage, as it was confirmed by the scanning electron microscopic images of bent specimens that were fixed in their bent state for inspection (Fig. 1c)."

Figure R1 (for reviewer's inspection only)

Comment 4. *The crystal length is 1 cm; what is the size of liquid nitrogen that was dropped continuously on a small area at one end?*

Response: A photograph of the experimental setup used for this experiment is shown below (note that the photo was taken under UV light for better contrast and visibility of the small crystalline sample). One end of the 1 cm length crystal was affixed to a hard paper by conductive adhesive, and the other end was left free in the air. The hard paper was tilted at a certain angle to avoid the exposure to the liquid nitrogen flow of the free end of the crystal. A zoomed-in picture of the crystal shows that the length of liquid nitrogen drop was about 0.5 mm. This photograph and the related details were added to the revised version of the Supplementary Information as Supplementary Figure 11. A video of the experiment of liquid nitrogen dropping was also recorded, and is provided in the revised version as a new supplementary video (Supplementary Movie 3).

Supplementary Figure 11

Comment 5: *Fig 3. Provide the scale for images.*

Response: We thank the Reviewer for catching this detail. The scale bar has been added to the revised Figure 3, and the original and the revised versions of the figure are provided below for Reviewer's perusal.

Original figure:

Revised figure:

Comment 6. How to ascertain that the crystal (length, 1.5 cm) temp is 77 K at one end and the other end is 230 K? What is the error in the reading? How was the temp measured in such a localized manner? Include a detailed scheme for the experimental set-up and the temperature sensors' specification.

Response: We thank the Reviewer for the insightful comment and the suggestion. The experimental setup is shown below. The cold end of the crystal was submerged into liquid nitrogen continuously, and, considering the small size of the crystal, we can safely assume that the temperature of the cold end is identical to that of the liquid nitrogen boiling temperature (77 K). The thermal probe for

temperature reading was placed next to the crystal's warmer end. The temperature of the warmer end of the crystal was expectedly reduced because it's positionally close to the liquid nitrogen drop (about 1 cm). When the flow rate of the liquid nitrogen was stabilized, the temperature of the warm end became stable at about 230 K, as confirmed by direct measurement.

Supplementary Figure 12

We would like to note that the experiment was repeated five times to ensure the reliability of result obtained by temperature measurement. Snapshots of the temperature display showing the temperature at the warm end are provided as new Supplementary Figure 13. As in the other experiments, the photos were captured under UV light for better visibility of the small crystalline samples. The temperature measured at the warm ends was 229.6 K, 229.5K, 229.1 K, 228.5 K and 229.4 K, which average to 229.2 K. A comment related to this point was added in the revised version of the main text (page 4, line 133). The temperature was measured by Miaoxin T10R-PT equipped PT100 sensor. This sensor has faster response speed and lager measuring range compared to traditional alcohol or mercury thermometers. The specifications of the temperature sensor are shown for reviewer's inspection in Table R1. The detailed information on the technical characteristics of the temperature sensor was added to the revised Supplementary Information.

Crystal 1

Crystal 2

Crystal 3

Crystal 4

Crystal 5

Supplementary Figure 13

Table R1 (for reviewer's inspection only)

Measuring range	Reading error	Resolution	Baud rate
73–473 K	± 0.3 K	0.1 K	115200 bit/s

Added text: “The temperature of the warm end was determined to be about 230 K (Supplementary Figures 12 and 13).”

Added text (Supporting Information): “The temperature was measured by MiaoXin T10R-PT temperature sensor with 0.1 K resolution (reading error: ± 0.3 K).”

Comment 7. Page 4: Though orange (cold) and green (warm) FL emitted at opposite crystal terminals, the observation of 580 nm peak, showing that only the orange light was transduced, needs to be reinterpreted. The spectral band shape and width are critical here. Fig. 3b spectrum is more symmetric (with a shoulder at about 520 nm), and Fig 3d is asymmetric (more intense on the left side). It looks like the low-intensity green signal is buried within the highly intense orange signal spectrum. This result shows that the orange and green signals are mixed inside the crystals, as evident from Fig. 3e spectral shoulder at intermediate temperatures. A similar interpretation also applies to the 550 nm signal.

Response: We thank the Reviewer for this important observation. While the crystal was excited by 355 nm laser, the optical waveguide is at active mode: the excited part of the crystal can be considered as an optical source (input) and the remaining portion of the crystal plays the role of an

optically transmissive medium. The PL spectrum at the excitation site (input) could be affected by low temperature, more specifically, the non-symmetric peak shape may be caused by the electronic structure change of the molecules at low temperature (77 K). The propagation of light in the waveguide does not change the PL spectrum. In order to further verify whether the green signal of the intermediate section was buried within the orange signal spectrum, we performed an additional experiment. The PL spectrum of crystals of millimeter size was recorded first, and then these crystals were immersed in liquid nitrogen for 2 minutes to make sure all the crystals were at 77 K, as shown in Supplementary Figure 14b (captured under UV light). The PL spectrum of these crystals at low temperature was then recorded. The PL spectra at 77 K and 298 K are shown in Supplementary Figure 14c.

Supplementary Figure 14

As can be concluded in the spectra shown in Supplementary Figure 14c, there is a clear difference of the PL spectrum due to the temperature difference. Comparing the 77 K (black curve) with the optical waveguide signal (red dashed line in Supplementary Figure 14d, excitation point temperature is 77 K, PL spectrum was collected at about 240 K), we conclude that the peak shapes are almost identical. This result indicates that the optical signal propagating in the waveguide is not affected during the propagation. In order to clarify this point, a comment was added to the main text (page 4,

line 135-140).

Added text: “We note that the optical waveguide is in active mode here. The photoluminescence (PL) spectra of crystals of millimeter size at 77 K and 298 K were also recorded. Comparing the PL spectrum at 77 K with the optical waveguide signal at 77 K, we conclude that the peak shapes are almost identical (Supplementary Fig. 14d). This result indicates that the optical signal propagating in the waveguide is not affected during its propagation.”

Reviewer #3

Comment: *This paper describes an interesting concept for fluorescence-based thermal sensing using elastic organic crystals. The reported crystalline material [(E)-1-(3,5-bis(trifluoromethyl)phenyl)-3-(1-ethyl-1H-indol-3-yl)prop-2-en-1-one (BPEPO)] exhibits an impressively linear correlation between the emission wavelength and fluorescence intensity between 77K and 277K. Density functional theory provides detail rationale for the observed behavior. This observation forms the foundation for a new thermal sensing mechanism and would be a valuable finding to share with the general scientific community of this journal. I recommend the authors consider the following points to improved their work:*

Comment 1. *The abstract mentions space exploration missions as a potential application for these fluorescence-based thermal sensors. Characterization about the lifetime of temperature sensing after repeated bending of one crystal would be valuable to connect back to this application. How is the sensitivity of BPEPO impacted after repeated bending? Has the lifetime of the flexibility of BPEPO been tested?*

Response: We thank the Reviewer for the careful reading and the valuable suggestions, which were addressed and contributed to further improvement of our manuscript. In order to respond to this comment, a crystal was selected to perform the variable temperature experiments, and the crystal optical waveguide signals at different temperatures were recorded after the crystal was bent 100, 300, 500, and 1000 times. These new results were added to the revised Supporting Information (Supplementary Fig. 17 and Supplementary Table 3), as shown below:

Supplementary Figure 17

Supplementary Table 3

Bend times	0	100	300	500	1000
$R^2 (I / I_{max})$	0.9916	0.9954	0.9927	0.9982	0.9977
$R^2 (\lambda_{max})$	0.9930	0.9943	0.9849	0.9960	0.9780
Sensitivity (I / I_{max})	-0.0019 /K	-0.0019 /K	-0.0020 /K	-0.0018 /K	-0.0021 /K
Sensitivity (λ_{max})	-0.17 nm /K	-0.17 nm /K	-0.17 nm /K	-0.17 nm /K	-0.17 nm /K

The above results illustrate that the sensitivity of the wavelength-vs-temperature curve remains unchanged, while the sensitivity of intensity-vs-temperature curve is reduced slightly after the crystal was bent 1000 times, although the sensitivity remains close to the average value. Based on these results, we conclude that the repeated bending (at least, on the order of cycles performed in this experiment) is unlikely to impact the sensitivity of the crystal. A comment to clarify this point was added to the text, and is provided below for the Reviewer's convenience (page 5, line162-165):

Added text: "The measurement was also performed after the crystal was repeatedly bent, and the results show that the sensitivity was practically unaffected even after the crystal was bent 1000 times (Supplementary Figure 17, Supplementary Table 3)."

We also tested the flexibility of BPEPO and inspected the surface of the samples by using scanning electron microscopy (SEM). Different single crystals were bent and unbent 100, 500, 1000, 3000, and 5000 times. SEM images of these crystals after bending were recorded, and were added to the Supplementary Information. They are provided for inspection below.

Supplementary Figure 4

The SEM images show that there are no obvious defects or damage on the surface of the crystals. A comment on this result has been added in the revised version of the main text (page 3, line 79-80). Although we were only able to perform the bending to a reasonable number of times (5000 times), we note that it is difficult to extrapolate this result or to estimate the upper limits of cycling to which the crystal would maintain its macroscopic integrity.

Added text: “No obvious defects could be observed on the crystal surface even after bending crystals for up to 5000 times (Supplementary Figure 4).”

Comment 2. *The introduction lacks background on the connection between fluorescence and organic crystals. This section would benefit from information about the molecular design of emissive crystals and aggregate-induced fluorescence. Adding in this information may also help contextualize the process that led to the discovery of BPEPO as an elastic emissive crystal, mentioned on page 2 line 66.*

Response: We thank the Reviewer for this suggestion, and we certainly agree with their comment. In response, new text and references referring to the fluorescence of organic crystals were added in the revised version (page 2, line 42-47 and References [14-18])

Added text: “Organic fluorescent crystals based on π -conjugated small organic molecules with photoluminescent properties have been reported.^[14] In crystalline condensed state, both the intermolecular interactions and the molecular arrangement are known to affect the luminescence.^{[15-}

^{17]}. Within that context, molecular organic crystals from π -conjugated molecule have been studied extensively as promising emissive materials due to the effects of molecular aggregation on the emission.^[18]”

References:

[14] Gierschner, J. et al. Luminescence in crystalline organic materials: From molecules to molecular solids. *Adv. Optical Mater.* **9**, 2002251 (2021).

[15] Gierschner, J., Varghese, S. & Park, S. Organic single crystal lasers: A Materials view. *Adv. Optical Mater.* **4**, 348–364 (2016).

[16] Yan, D. & Evans, D. Molecular crystalline materials with tunable luminescent properties: From polymorphs to multi-component solids. *Mater. Horiz.*, **1**, 46–57 (2014).

[17] Guerrini, M., Calzolari, A. & Corni, S. Solid-state effects on the optical excitation of push–pull molecular J-Aggregates by First-Principles simulations. *ACS Omega.* **3**, 10481–10486 (2018).

[18] Li, Q. & Li, Z. The strong light-emission materials in the aggregated state: What happens from a single molecule to the collective group. *Adv. Sci.* **4**, 1600484 (2017).

Comment 3. *Page 2, lines 52-55, the structure of this sentence is difficult to read.*

Response: We thank the Reviewer for the comment. The sentence has been revised (page 2, line 58-61) as shown below and we hope it is clearer in the revised form:

Original text: “In line with the increasing demand for resilient cryogenic materials, we sought out to developing light-weight, mechanically compliant materials that can be specifically used for sensing or optical transduction of information at low temperatures; all of these assets are contained within a single material that we report here.”

Revised text: “In line with the increasing demand for resilient cryogenic materials, we sought out to developing light-weight, flexible materials that can be used for sensing or optical information transduction at low temperatures. The material that we report here combines all of these assets.”

Comment 4. *Page 3, line 75, was the reported bulk elastic modulus measured at room temperature? Was the bulk elastic modulus measured at lower temperatures?*

Response: We thank the Reviewer for the important remark. The elastic modulus was measured at room temperature, and this information has been added in the revised main text (page 3, line 84-85). Unfortunately, we were not able to access the elastic modulus at lower temperature due to technical limitations with the equipment available to us.

Original text: “Their bulk elastic modulus was found to be 1.6 GPa by tensile measurements (Fig 1e).”

Revised text: “Their bulk elastic modulus at 298 K was found to be 1.6 GPa by tensile measurements (Fig 1e).”

Comment 5. *The sensitivity of BPEPO seems to be reported as the slope of the intensity vs temperature curve (Page 3, lines 86 and 90). How does this compare to the sensitivity of other thermal sensors?*

Response: In order to respond to this comment, we inspected the available related literature on thermal sensors. The usual sensitivity of a thermal sensor based on metal or metal alloy is between 0.01 mV/K and 0.06 mV/K (ref. [43]), and the usual sensitivity of a thermal sensor based on silicon thermodiodes is between 1 mV/K and 2 mV/K (ref. [44]). These temperature sensors are realized by using different resistance at different temperatures. Some other organic materials with temperature-dependent fluorescence can also respond to temperature changes by changes in fluorescence, and their sensitivity (intensity vs. temperature) is between 0.016 1/K and 0.05 1/K (refs. [45,46]). While the sensitivity (intensity vs. temperature) of 0.0022 1/K of BPEPO can not be directly compared with those of metal or silicon thermal sensors because of different units, we note that the sensitivity (maximum emission wavelength vs. temperature, 0.21 1/K) is higher than those of other organic materials with temperature-dependent fluorescence. A comment on this was added to the revised version of the main text (page 3, lines 99-105).

Original text: “The change of emission intensity with temperature is also linear within a certain temperature range ($I/I_{\max} = 1.14 + -0.0022 T/K$, $R^2 = 0.9711$; Fig. 2d). This linear and strong response of the fluorescence of BPEPO with temperature favors this material as a temperature-sensing medium.”

Revised text: “The change of emission intensity with temperature is also linear within a certain temperature range ($I/I_{\max} = -0.0022 T/K + 1.14$, $R^2 = 0.9711$; Fig. 2d). This linear and strong response of the fluorescence of compound **1** with temperature favors this material as a temperature-sensing medium. The lowest detectable temperature of compound **1** is lower than that of some metal sensors,^[43,44] however, the sensitivity (maximum emission wavelength vs. temperature) is higher than other organic material with temperature-dependent fluorescence.^{[45,46]”}

Added references:

[43] Reverter, F. A Tutorial on thermal sensors in the 200th anniversary of the seebeck effect. *IEEE Sensors Journal*, **21**, 22122–22132 (2021).

[44] Mansoor, M., Haneef, I., Akhtar, S., Luca, A. & Udrea, F. Silicon diode temperature sensors-a review of applications. *Sensors and Actuators A*. **232**, 63–74 (2015).

[45] Liu, D., Sun, Z., Zhao, Z., Peng, Q. & Zhao, C. 1,1'-Binaphthyl consisting of two donor–p–acceptor subunits: A general skeleton for temperature-dependent dual fluorescence. *Chem. Eur. J.* **25**, 10179–10187 (2019).

[46] Sun, Z. et al. 2,2'-Diamino-6,6'-diboryl-1,1'-binaphthyl: A versatile building block for temperature-dependent dual fluorescence and switchable circularly polarized luminescence. *Angew.*

Comment 6. Page 4, lines 124-125, why is the output not affected when the intermediate section of the crystal is at a lower temperature? Is this common for crystal waveguides?

Response: We thank the Reviewer for this important comment. When the crystal was excited by laser (355 nm), the optical waveguide is at active mode (*Angew. Chem. Int. Ed.* 57, 17254–17258 (2018)). While the crystal was excited by 355 nm laser, the optical waveguide is working in active mode: the excited part of the crystal can be considered as an optical source (input) and the remaining portion of the crystal plays the role of optical transmission medium. The PL spectrum at the excitation site (input) could be affected by low temperature; the optical signal propagating in the waveguide is not affected during the propagation process in this experiment.

Following the suggestion from this and another Reviewer, we performed additional experiments to clarify this. Crystals that were shorter than 5 mm were immersed in liquid nitrogen for 2 minutes to make sure all the crystals were at 77 K, and their PL spectrum was recorded. Compared to the optical waveguide signal of a 1.5 cm-long crystal at 77 K, the peak shapes were identical (Supplementary Figure 14d). This result indicates that the crystal optical waveguide signal is consistent with the PL spectrum of the excited part. The low temperature does not affect the crystal's ability to transmit light (*Angew. Chem. Int. Ed.* 59, 23117-23121 (2020)), as is common for crystal waveguides. Analogous results on two other crystals, compound 2 (*Angew. Chem. Int. Ed.* 2022,61, e2022039) and compound 3 (*Angew. Chem. Int. Ed.* 58, 19081–19086 (2019)) that have been reported before are in line with the results on compound 1 (BPEPO). A comment and the results were added to the revised version of the Supplementary information, and they are provided for the Reviewer's convenience below:

Supplementary Figure 14d

Crystal 2

Crystal 3

Supplementary Figure 15

The Supplementary Figure 15 shows that the fluorescence of both compound **2** (“Crystal 2”) and compound **3** (“Crystal 3”) are blue-shifted at 77 K. Similar to the BPEPO crystal that we report here, the output signals are not affected when the intermediate section of the crystal is at lower temperature. A comment to clarify this point was added to the revised version of the main text (page 4, line 135-141):

Added text: “We note that the optical waveguide is in active mode here. The PL spectra of crystals of millimeter size at 77 K and 298 K were also recorded. Comparing the PL spectrum at 77 K with the optical waveguide signal at 77 K, we conclude that the peak shapes are almost identical (Supplementary Figure 14). This result indicates that the optical signal propagating in the waveguide is not affected during its propagation. Other two crystals that have been reported earlier (compounds **2** and **3**) were also investigated to confirm this property of crystals (Supplementary Fig. 15).”

Comment 7. Page 6, lines 181-182, the claim about the non-radiative and radiative pathways at low temperature could be strengthened by calculating the non-radiative and radiative rate constants at 77K and 277K from the fluorescence lifetime data.

Response: We thank the Reviewer for the valuable suggestion. According to the formulas for radiative rate constants $K_r = Y_f/\tau$ and non-radiative rate constants $K_{nr} = (1-Y_f)/\tau$, the fluorescence quantum yield of the compound could be calculated. Very unfortunately, we were unable to obtain the quantum yield at 277 K due to technical limitations of our experimental setup. We therefore used another instrument to determine the quantum yield at 77 K and 298 K and to calculate K_r and K_{nr} . As expected, the results were slightly different from the quantum yield in the text, and the revised text is provided below (page 3, lines 80-84):

Original text: “The crystals absorb light with a maximum at 415 nm, and emit bright green fluorescence with a maximum emission at about 540 nm and a quantum yield of 0.20 (Fig 1d).”

Revised text: “The crystals absorb light with a maximum at 415 nm, and emit bright green fluorescence with a maximum emission at about 540 nm and quantum yield of 0.16 at 298 K (Fig 1d). The fluorescence quantum yield is 0.30 at 77 K, which is higher compared to room temperature due to the decreased non-radiative rate at low temperature (Supplementary Table 1).”

Supplementary Table 1

Temperature	Lifetime, τ	PLQY, Y_f^a	Radiative rate constant, K_r	Non-radiative rate constant, K_{nr}
77 K	17.99 ns	0.30	1.67×10^7	3.89×10^7
298 K	6.13 ns	0.16	2.61×10^7	1.37×10^8

^aPLQY stands for photoluminescence quantum yield.

Comment 8. Page 6, lines 199-200, does the mechanism that allows for the flexibility of BPEPO also promote the non-radiative pathways at room temperature? Is there a reduction of flexibility that corresponds to emission enhancement?

Response: To respond to this comment, we determined the quantum yield and fluorescence lifetime of bent crystals at 298 K (the emission wavelength in the fluorescence lifetime test is 540 nm), and we calculated the radiative rate constant and non-radiative rate constant. The related discussion was added to the revised version of the Supporting Information (Supplementary Figure 6 and Supplementary Table 2), and is also provided below for convenience:

Supplementary Figure 6

Supplementary Table 2

Shape of the crystal	Lifetime, τ	PLQY, Y_f^a	Radiative rate constant, K_r	Non-radiative rate constant, K_{nr}
straight	6.13 ns	0.16	2.61×10^7	1.37×10^8
bent	5.95 ns	0.16	2.69×10^7	1.41×10^8

^aPLQY stands for photoluminescence quantum yield.

As can be concluded from the above results, the K_{nr} of bent crystals is almost equal to the K_{nr} of straight crystals, and their quantum yields are also almost identical. This result can be taken to demonstrate that the bending does not promote the non-radiative pathways at room temperature, and it does not correspond to emission enhancement either. A related discussion was added to the revised text (page 4, lines 109-111).

Added text: “The lifetime and quantum yield of bent crystals were also recorded and indicate that the bending does not have any significant effect on non-radiative pathways (Supplementary Figure 6, Supplementary Table 2)”.

Comment 9. Does the fluorescent emission of BPEPO change at warm temperature (>277K)? If so, is this material as sensitive to warm temperature changes as it is towards cold temperature changes?

Response: We thank the Reviewer for this important comment. To respond to the comment, the fluorescence spectra from 297 K to 457 K were recorded, and the plots of I/I_{max} vs temperature and λ_{max} vs temperature are shown below:

Figure R2 (for reviewer's inspection only)

As it can be concluded from these results, the fluorescence intensity decreases as the temperature increases, and the dependence is linear in the temperature range from 297 K to 457 K ($I/I_{max} = -0.0021 T/K + 1.64$, $R^2 = 0.9927$). The sensitivity is almost same as the crystal in the temperature range from 77 K to 277 K, however, the emission maxima wavelength is hardly changed when the temperature is above 277 K.

Comment 10. *In Figure 1, it's difficult to make out the details in images h. and i.*

Response: We thank the Reviewer for noting this. The crystal was completely immersed in the liquid nitrogen, and the liquid nitrogen bubbling was recorded. The photos were replaced with new version, and the revised Figure 1 is provided below.

Original Figure 1:

Revised Figure 1:

Comment 11. The caption of Figure 1 mentions image b. depicts BPEPO under daylight. Are the other images also captured under daylight?

Response: The image b was captured under daylight, but the other photos (panels f–m) were all captured under UV light. The caption of Figure 1 was revised to highlight this difference.

Original Figure 1 caption: “Structure, elasticity and fluorescence of BPEPO. a Chemical structure of BPEPO. b Photo of BPEPO crystals under daylight. c SEM images of a bent crystal. d Absorption and emission spectrum of a BPEPO crystal. e Stress-strain curve of the crystal. f–g Photos showing crystal bending in air. h–i Photos showing crystal bending in liquid nitrogen. j–o Change in emission color of the crystals from 298 K to 77 K.”

Revised Figure 1 caption: “Structure, elasticity and fluorescence of compound 1. a Chemical structure of compound 1. b Photograph of crystals of compound 1 under daylight. c SEM images of a bent crystal. d Absorption and emission spectrum of crystals of compound 1. e Stress-strain curve of the crystal. f–g Photographs showing crystal bending in air. h–i Photographs showing crystal bending in liquid nitrogen. j–o Change in emission color of the crystals from 298 K to 77 K (the images shown in panels f–m were recorded under UV light for better contrast).”

Comment 12. *In the caption of Figure 2, the description of the locations on the crystal is unclear under part e, f.*

Response: We thank the Reviewer for catching this. The caption has been revised to make the description clear.

Original caption of Figure 2e, f: “Emission spectra measured at one end of the crystal at various distances (0–5 mm) between the end and the excitation position at 298 K and 77 K, respectively.”

Revised caption of Figure 2e, f: “e,f Emission spectra measured at one end of the crystal at 298 K (e) and 77 K (f). The values 0–5 mm represent the distance from the excitation site to the point of measurement. The excitation was performed with 355 nm light.”

Comment 13. *In Figure 3, part a and c, what is the dashed grey box denoting?*

Response: The dashed grey box denotes the area where the liquid nitrogen was dropped. The caption of Figure 3a has been revised to reflect this.

Original caption of Figure 3a: “Schematic diagram and real picture of a crystal waveguide at low temperature.”

Revised caption of Figure 3a: “Schematic diagram and real picture of a crystal waveguide at low temperature. The grey dashed box denotes the area where the liquid nitrogen was dropped (77 K).”

Comment 14. *In Figure 4, part b, are the colors of the S1, S2, and S3 levels meant to communicate information about emission? If so, that should be mentioned in the figure caption. If not, consider using greyscale for these levels.*

Response: We thank the Reviewer for their comment. The colors of the S1, S2 and S3 levels do not relate to the emission. A greyscale coloring was used in the revised figure.

Original Figure 4:

Revised Figure 4:

Comment 15. Supporting Information should include cartesian coordinate files of optimized geometry for all structures.

Response: The coordinates of the optimized geometry for all structures have been added to the Supporting Information (Supplementary Table 5).

Comment 16. *The ramp rate used for DSC measurements should be reported in Figure S4 caption or in General Information section of the Supporting Information.*

Response: The ramp rate of the DSC measurement was 20 °C/min and it has been added to the caption of Supplementary Figure 3.

Original Supplementary Figure 3 caption: Differential thermal calorimetric analysis of the crystals.

Revised Supplementary Figure 3 caption: Differential Scanning Calorimetric (DSC) analysis of crystals of compound **1** recorded at heating/cooling rate of 20 °C/min.

Comment 17. *The method used to calculate the quantum yield should be detailed in the General Information section of the Supporting Information.*

Response: The fluorescence quantum yield was tested by Hamamatsu Quantaaurus quantum yield spectrometer. The method used to calculate fluorescence quantum yield is as follows. When no sample was placed, light source (365 nm) was used to excite the blank sample pool to obtain the spectrum. Then the sample was placed in the sample pool to record another fluorescence spectrum. The integral difference between the two spectral excitation range was denoted as S1 and the integral sample emission range was denoted as S2. The quantum yield was calculated as ratio of S2 to S1. The low temperature environment in the quantum yield test is provided by liquid nitrogen. This part has been added to the Supporting Information.

Added General Information: The fluorescence quantum yields were determined by using a Hamamatsu Quantaaurus quantum yield spectrometer. The light source (365 nm) was first used to excite a blank sample to obtain the background spectrum, and then the sample was placed in the sample pool to record its fluorescence spectrum. The integral difference between the two spectral excitation range was denoted as S1 and the integral sample emission range was denoted as S2. The quantum yield was calculated as ratio of S2 to S1. The low-temperature for the quantum yield measurement was obtained by using liquid nitrogen.

REVIEWERS' COMMENTS

Reviewer #1 (Remarks to the Author):

The authors have done an excellent job in responding to the referees' comments. The paper is acceptable for publication.

Reviewer #2 (Remarks to the Author):

The authors have addressed the queries and suggestions in the revised version. Therefore, I recommend the publication of this paper.

Reviewer #3 (Remarks to the Author):

The authors have addressed my comments. I believe this paper is now suitable for publication.

RESPONSE TO THE REVIEWERS' AND TECHNICAL COMMENTS

Manuscript:

In the PDF version of this response, the reviewers' comments are written in black italic font, our responses to the comments are written in blue regular font, and the changes made to the main text are written in red regular font.

Reviewer #1 (Remarks to the Author):

Comment: The authors have done an excellent job in responding to the referees' comments. The paper is acceptable for publication.

Response to the comment: We thank the Reviewer for their valuable input and we are glad that we have addressed the Reviewer's comments to their satisfaction.

Reviewer #2 (Remarks to the Author):

Comment: The authors have addressed the queries and suggestions in the revised version. Therefore, I recommend the publication of this paper.

Response to the comment: We thank the Reviewer for their valuable input and we are glad that we have addressed the Reviewer's comments to their satisfaction.

Reviewer #3 (Remarks to the Author):

Comment: The authors have addressed my comments. I believe this paper is now suitable for publication.

Response to the comment: We thank the Reviewer for their valuable input and we are glad that we have addressed the Reviewer's comments to their satisfaction.